# Synergies between urban heat island and heat waves in Seoul: The role of wind speed and land use characteristics

Jack Ngarambe[1], Jacques Nganyiyimana[1], Inhan Kim[2], Mat Santamouris[1,3], Geun Young Yun [1]*

1 Department of Architectural Engineering, Kyung Hee University, Giheung-gu, Yongin-si, Gyeongi-do, Republic of Korea, 2 Department of Architecture, Kyung Hee University, Giheung-gu, Yongin-si, Gyeongi-do, Republic of Korea, 3 Faculty of Built Environment, University of New South Wales, Sydney, New South Wales, Australia

* gyyun@khu.ac.kr

**Data Availability Statement:** All relevant data are within the manuscript and its Supporting Information files.

**Funding:** This work was supported by Korea Institute of Energy Technology Evaluation and

## Abstract

The effects of heat waves (HW) are more pronounced in urban areas than in rural areas due to the additive effect of the urban heat island (UHI) phenomenon. However, the synergies between UHI and HW are still an open scientific question and have only been quantified for a few metropolitan cities. In the current study, we explore the synergies between UHI and HW in Seoul city. We consider summertime data from two non-consecutive years (i.e., 2012 and 2016) and ten automatic weather stations. Our results show that UHI is more intense during HW periods than non-heat wave (NHW) periods (i.e., normal summer background conditions), with a maximum UHI difference of 3.30°C and 4.50°C, between HW and NHW periods, in 2012 and 2016 respectively. Our results also show substantial variations in the synergies between UHI and HW due to land use characteristics and synoptic weather conditions; the synergies were relatively more intense in densely built areas and under low wind speed conditions. Our results contribute to our understanding of thermal risks posed by HW in urban areas and, subsequently, the health risks on urban populations. Moreover, they are of significant importance to emergency relief providers as a resource allocation guideline, for instance, regarding which areas and time of the day to prioritize during HW periods in Seoul.

## 1. Introduction

The world has seen recent increases in temperature that are, arguably, attributable to climate variability and increased greenhouse gas emissions that intensify radiative forcing [1]. This increase in global temperature results in adverse effects on the ecosystem, food security, water supply, human health, and well-being. One of the consequences of increased temperatures with definitive and detrimental effects on human health is the occurrence of heat waves (HW). HWs are defined as episodes of prolonged high temperatures resulting in increased thermal stress and often linked to heat-related health conditions (e.g., pulmonary heart diseases and

Planning (KETEP) grant funded by the Korean government (MOTIE) (20192020101170, Development of Energy-Recovery Ventilators Equipped with Air Filters for Fine Dust Removal). This research was supported by a grant (20AUDP-B127891-04) from the Architecture & Urban Development Research Program funded by the Ministry of Land, Infrastructure and Transport of the Korean government. The funders had no role in study design, data collection and analysis, decision to publish, or preparation of the manuscript.

**Competing interests:** The authors have declared that no competing interests exist.

increased occurrence of strokes) especially among vulnerable populations (i.e., the elderly, the unhealthy and those from societies that experience energy poverty [2–5].

HWs have also been reported to significantly contribute to the incessant number of heat-related mortalities around the world; the 2003 European HWs, which resulted in approximately 70,000 deaths, is a good example of the extreme fatal risk posed by HWs [6]. Relatively less severe but equally high numbers of HW-related mortalities have been reported in the United States [7], Russia [8] and Korea [9]. Moreover, studies have shown that global warming-related increase in temperature is likely to exacerbate both the intensity and frequency of HW in many parts of the world. Also, through dynamic modeling and simulation studies, there is strong evidence that the occurrence, severity and duration of HW periods are also likely to increase in the near future [10, 11].

Besides the global warming-related effects that are often boundary-less and experienced on a much broader scale, urban areas specifically tend to experience even higher temperatures than rural/suburban areas due to the additive effect of urban heat island (UHI). The UHI phenomenon is widely documented in the literature and refers to the often elevated temperatures in urbanized areas relative to rural/suburban areas [12–15].

UHI is often a result of drastic land-use change seen in urbanized areas, usually in efforts to accommodate the incessant increase in urban populations (i.e., infrastructure development). Such changes, through complex atmospheric mechanisms, distort the natural state of urban climates, making them significantly warmer than rural climates, especially during nocturnal and winter times. Some of the factors that contribute to the occurrence of UHI include reduced evaporative cooling and increased thermal storage capacities by artificial construction materials prevalent in urban areas [16], distorted wind flow mechanisms as a result of tall and densely constructed buildings [17], as well as anthropogenic activities such as those related to occupant behavior and particularly the space cooling/heating of buildings [18–20].

Consequently, HWs in urban areas are likely to be more severe than those in rural areas due to the additive effects of UHI-related warming. For instance, Yang et al. [21] report a 1.9°C difference in surface temperature between American metropolitan cities and their rural counterparts during HW periods. Similarly, Tan et al. [22] observed higher degrees of warming in Shanghai relative to its surrounding area during HW periods. Moreover, heat-related mortalities during HW periods have been observed more in urban areas than in rural areas [23, 24]. In essence, therefore, urban dwellers are more vulnerable to heat-stress-related health conditions than rural dwellers as they are likely to combat HWS and UHI's synergistic effects.

The synergistic interactions between UHI and HW have been established in several previous studies. For instance, an amplified UHI of up to 1.5°C was reported in Beijing during HW periods compared to NHW periods [25]. Also, Founda and Santamouris [26] studied synergies between UHI and HW periods during summertime in Greece. They found increased UHI magnitudes of up to 3.5°C during HW periods compared to ordinary summer conditions. Similarly, Khan et al. [27] explored UHI magnitudes during HW and NHW periods in Australia. They found UHI levels in HW periods higher than those in NHW periods by a magnitude of 8°C. The amplified UHI during HW periods is also empirically tied to increased heat-related mortalities and morbidities. Tan et al. [22] observed a higher number of HWs and a subsequent higher number of HW-related deaths in Shanghai than in neighboring rural areas. Heavside et al. [28] also report a 50% increase in HW-related mortalities attributable to UHI in the west midlands of the United Kingdom.

While the synergistic interactions between UHI and HWs have been studied, to an extent, in some large cities around the world which are discussed above [26–29], it is essential to remark that the synergies between UHI and HW are likely to be influenced by factors (e.g., synoptic weather, topography, population density, land use) specific to a given region/city. As

such, the synergies between UHI and HW within a given area are distinctive rather than general and must be quantified for individual areas of interest. To that end, we attempt to explore the potential synergistic interactions between UHI and HW in Seoul, a large metropolitan city with a dense population, advanced urbanization, which inherently suffers from the urban warming-related problems discussed above.

Moreover, advective flux, as well as latent and sensible heat flux, are seen as the primary causes of the interactions between UHI and HW [30]. Consequently, we hypothesize that different local climatic zones (LCZ), as defined by Stewart and Oke [31], could influence the said interactions between local UHI and HW given the possible difference in convective flux and wind flow patterns among the LCZ categories. Also, the advective cooling resulting from prevailing winds naturally reduces local warming. This heat-mitigative effect of wind speeds has been illustrated in areas of various morphologies and building structures ranging from low-rise [32] to open mid-rise gridiron precincts [33]. It has also been previously shown that above certain wind speed thresholds, UHI is fully diminished / gets close to zero [34]. We thus study the difference in UHI magnitudes between HW and NHW periods under different wind conditions.

The ambient temperature data used in our study were collected from 10 automatic weather stations (herein referred to as stations) located in Seoul city in the summer of 2012 and 2016. The current study sheds light on the interactive processes between UHI and HW and highlights major factors that fuel such interactions in Seoul city. To the best of our knowledge, this is the first time such a study is being conducted in Seoul city. It thus serves to contribute to scientific discussions on HW-related policies and mitigation measures in Seoul city.

## 2. Methods

### 2.1. Study area

The target area for our study is Seoul, South Korea's capital city and its largest metropolitan area. Seoul is located at a longitude of 126.59˚ E and latitude of 37.34˚ N, which ultimately falls in the central-western part of the Korean peninsula. It experiences a humid continental and subtropical climate with relatively high annual precipitation levels owing to the summer monsoons that are prevalent in East Asia and a mid-range, mean annual temperature of approximately 24˚C [35]. Over the last 40 years, South Korea's level of urbanization has increased at an alarming rate from 18.4% in the early fifties to 86.2% in the early two thousand and has continued to grow ever since [36]. This rapid increase in urbanization has exposed Seoul to drastic increases in population: it currently harbors 21% of the Korea's general population [37]. Simultaneously, the infrastructure, such as buildings and roads, has seen a significant increase as a means to cater to the increasing urban population, making Seoul a densely built area. Consequently, the increase in anthropogenic heat as a result of the increased population and the adverse changes in land use brought upon by the increased number of infrastructure substantially modulate Seoul's natural climate, making it susceptible to human-induced temperature changes. For instance, the KMA estimates that the average summer temperature in Korea has increased by 0.7˚C over the last 30 years [35]. Strong surface and tropospheric warming around the Korean peninsula are also supported by aerial observations [38]. Seoul is therefore vulnerable to HWs, as already seen in the recent past [39], making it an ideal area to explore the potential synergies between UHI and HWs.

### 2.2. Data collection and local climatic zones (LCZ)

The temperature and wind speed data used in our study are based on ground observations from ten stations in Seoul city collected in the summer months (i.e., May, June, July, August,

September) of two non-consecutive years (i.e., 2012 and 2016). The stations use metallic systems with thin films to sense and record hourly temperatures ranging from—40°C to 60°C with an accuracy of ± 0.3°C. To monitor wind speeds, the stations use ultrasonic sensors that measure wind speeds between 0 m/s– 70 m/s with an accuracy of ± 0.5 m/s. Furthermore, the stations are located in different areas characterized by distinct and individual physical features that may significantly modify local climatic conditions (i.e., advective heat flux and wind flow mechanisms). To study the potential influence of the structural and land cover differences among the stations on the interactions between UHI and HW therefore, we categorized our stations into four different groups following Stewart and Oke [31]; while they describe a wide range of local climatic zones, our stations were identified to fall into only five of the categories. Fig 1 shows the considered stations, while Table 1 shows the LCZ grouping for each station.

## 2.3. Identifying heat wave (HW) periods and quantifying urban heat island (UHI)

Currently, there is no consensus on the scientific definition of HW. Various definitions and thresholds have been adopted in the past primarily based on the duration of the HW and the spatial extension of the HW as well as various bioclimatic indices used to define extreme temperature conditions [40–42]. In many instances, however, HW episodes are defined as a particular number of consecutive days when the maximum temperature ($T_{max}$) surpasses a certain

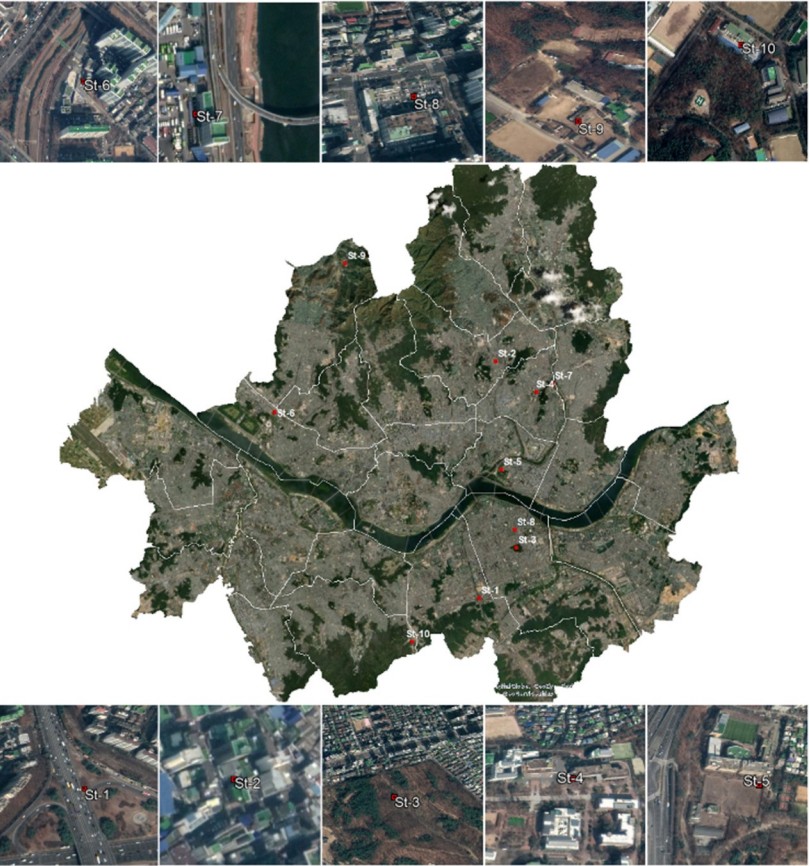

**Fig 1. Geographical locations of the station within Seoul city's boundaries (Contains information from OpenStreetMap and OpenStreetMap Foundation, which is made available under the Open Database License).**

**Table 1. Geographical locations of the considered stations.**

| Station | Latitude [˚N] | Longitude [˚E] | Local Climate zone | Building surface fraction [%] | Height of roughness elements [m] |
|---|---|---|---|---|---|
| ST-1 | 37.48 | 127.02 | LCZ 1 | 43.60 | 27.33 |
| ST-2 | 37.59 | 127.03 | LCZ 1 | 40.50 | 28.10 |
| ST-3 | 37.50 | 127.04 | LCZ 2 | 44.60 | 20.25 |
| ST-4 | 37.58 | 127.06 | LCZ 2 | 41.80 | 18.88 |
| ST-5 | 37.54 | 127.03 | LCZ 4 | 23.00 | 26.76 |
| ST-6 | 37.57 | 126.90 | LCZ 4 | 23.20 | 26.77 |
| ST-7 | 37.58 | 127.06 | LCZ 5 | 20.30 | 20.17 |
| ST-8 | 37.51 | 127.04 | LCZ 5 | 37.60 | 21.71 |
| ST-9 | 37.64 | 126.94 | LCZ A | 4.40 | 27.50 |
| ST-10 | 37.46 | 126.98 | LCZ A | 7.60 | 15.05 |

threshold. This threshold can be defined as an absolute temperature value or relative temperature value considering mean or maximum daily temperature distributions within a given period (e.g., 95th, 97th percentiles). Both of these approaches are widely adopted in the literature [27, 43]. In this study, we define a HW episode as a period of three or more consecutive days when $T_{max} > 33˚C$. This definition is dictated by the Korean Meteorological Agency (KMA) and has been employed in numerous studies that explore the occurrence of HWs in South Korea [9, 43, 44]. The adopted threshold corresponds to the 90th percentile of maximum daily summertime in Korea.

UHI is quantified using the UHI intensity (UHII) metric. UHII is defined as the difference in ambient temperature typically measured at the mean height of the boundary layer between an urban and rural area [31]. Although this urban-rural temperature difference approach provides a simple, systematic method to quantify UHI, the rural reference area must be selected appropriately to avoid inaccurate UHI estimates. As an example, the WMA (World Meteorological Agency) provides elaborate guidelines to help researchers select appropriate reference rural areas. Typically, rural reference areas should be located in relatively flat terrain surrounded by natural settings without impervious surfaces and densely constructed buildings extending above ground level commonly found in urbanized areas. Following such guidelines, we chose the Neuggok station located in the northern area of Seoul at a longitude of 126.47˚E and latitude of 37.37˚N. The Neuggok weather station has also been used in several UHI-related studies in Seoul [45–47]. UHII was therefore calculated as $T_{urban} - T_{rural}$ with $T_{urban}$ representing air temperature values observed at the earlier mentioned ten stations and $T_{rural}$ representing air temperature values recorded at the Neuggok station. Additionally, to gauge the difference in UHI amplitude between HW and NHW periods ($\Delta$UHII), we subtracted UHII at a time $x$ during NHW periods from UHI at a time $x$ during HW periods.

## 2.4. Data analysis

Our dataset consisted of numerous erroneous recordings and missing data typical of large climate datasets. Such recordings are, in many cases, a result of faulty equipment. For instance, the same temperature values observed over long periods are likely to be inaccurate recordings. Similarly, large temperature variations in relatively short time intervals are also likely to be wrong recordings often attributable to equipment failure. Estevez et al. [48], give a detailed discussion on the methods used to deal with such inaccurate recordings but often at the expense of reduced data size and efficiency. Consequently, to preserve the size of our data set, which can be particularly important in instances of small datasets, we first converted the said inaccurate recordings into missing values. We then employed linear interpolation techniques to

impute the missing values. Linear interpolation is a data imputation technique that assumes a linear trend within a variable and subsequently estimates missing data through learning the patterns in adjacent datasets; this method has been adopted and validated by several previous studies [49, 50]. Furthermore, to explore the potential effect of wind speed on HW episodes, our wind speed data were grouped into three categories of varying wind speeds using K-means clustering techniques [51]. This resulted in three clusters classified as low wind speeds (0 m/s–1.52 m/s), medium wind speeds (1.53 m/s– 3.2 m/s) and high wind speeds (3.2 m/s—13.7 m/s). To assess the mean UHII differences among different wind speed categories and LCZs, we employed analysis of variance (ANOVA) tests; the statistical significance level was determined at p < 0.001.

## 3. Results

Table 2 shows the total number of data entries analyzed for the two considered years (i.e., 2012 and 2016). Upon the completion of the data analysis steps discussed in section 2.4, the total number of data entries were 36720. For the entire period, the maximum UHII values were 7.5˚C and 9.2˚C recorded in 2012 and 2016, respectively. The mean annual values of UHII were 0.91˚C for the year 2012 and 1.72 for the year 2016. The maximum annual recorded wind speed values were 13.7 m/s and 10.1 m/s in 2012 and 2016, respectively. Based on the criteria discussed in section 2.3, there were a substantial number of HW periods in both years. A total of 13 HW periods and 38 HW periods were observed in 2012 and 2016, respectively. The higher number of HW in 2016 than in 2012 also corresponded to a higher mean UHII value in 2016 than in 2012.

### 3.1. UHI magnitude between HW and NHW periods

UHII calculated for each station ranged between -0.73˚C and 3.78˚C for HW periods and -1.15˚C and 0.89˚C for NHW periods in 2012. In 2016, the ranges were between -1.37˚C and 4.43˚C for HW periods and -0.81˚C and 3.67˚C for NHW periods. The results indicate amplified UHI levels during HW periods than NHW periods. While UHII was higher during HW periods at all target stations, its intensity varied substantially among the considered stations. Generally, (S1–S10 Figs), the largest UHII variations between HW and NHW periods were observed at ST-1 (i.e., 1. 56˚C in 2012, 1.05˚C in 2016), ST-2 (i.e., 1.57˚C in 2012, 0.64˚C in 2016), ST-3 (i.e., 1.53˚C in 2012, 0.76˚C in 2016) and ST-4 (i.e., 1.73˚C in 2012, 0.66˚C in 2016). These four stations belong to LCZ1 and LCZ2, which are particularly associated with high building densities. Surprisingly, however, stations categorized in LCZ A, which is characterized by low building densities, also experienced high UHII variations between HW and NHW periods compared to stations from the other LCZs.

To further explore the potential synergistic interactions between HW and UHI, we compared the diurnal difference in UHII (ΔUHII) between HW periods and NHW periods for the

**Table 2. Descriptive statistics of the data used in the study.**

| Year | Variable | N | Minimum | Maximum | Mean | Standard deviation |
|------|----------|---|---------|---------|------|--------------------|
| 2012 | UHII [˚C] | 36720 | - 7.50 | 7.50 | 0.91 | 1.66 |
|      | Wind speed [m/s] | 36720 | 0.00 | 13.70 | 1.48 | 1.02 |
|      | Number of HWs | | | 13 | | |
| 2016 | UHII [˚C] | 36720 | -26.50 | 9.20 | 1.72 | 1.75 |
|      | Wind speed [m/s] | 36720 | 0.00 | 10.10 | 1.39 | 1.01 |
|      | Number of HWs | | | 38 | | |

ten candidate stations combined and the separate years of interest (i.e., 2012 and 2016). ΔUHII between HW periods and NHW periods was obtained through a simple arithmetic approach that included subtracting the prevailing UHII at a particular time and stations during the NHW period from the prevailing UHII conditions at the same time and stations during HW periods. ΔUHII was positive at all hours of the day and for all ten candidate stations indicating that UHII values during the HW period were always higher than those during the NHW period. The average of all maximum ΔUHII values recorded at all stations was 1.02°C and 0.9°C in 2012 and 2016, respectively. The absolute maximum ΔUHII values recorded, considering all stations were 3.30°C and 4.50°C in 2012 and 2016, respectively. While ΔUHII was positive for all stations at all hours of the day, it was also more intense at certain stations than the others, indicating a potential diurnal variation in the synergistic interactions between UHI and HW. The peak ΔUHII values were typically observed in the early afternoons to late nights and the lowest ΔUHII values were generally observed in the early mornings (i.e., 6 am– 9 am), suggesting intensified synergistic interaction between UHI and HW in a particular time frame. Fig 2 shows the average and maximum ΔUHII calculated, considering all the ten candidate stations.

Furthermore, we assessed the differences in absolute humidity between HW and NHW periods for our urban and reference stations. Fig 3 shows the absolute humidity values averaged for each hour during the HW and NHW periods. As seen in the figure, absolute humidity was substantially higher during HW periods than NHW periods. The higher absolute humidity values during HW periods than NHW periods were observed at both the urban stations and the reference station. However, the magnitude was particularly higher at the rural station than the urban stations. For example, in 2012, the highest absolute humidity value recorded during

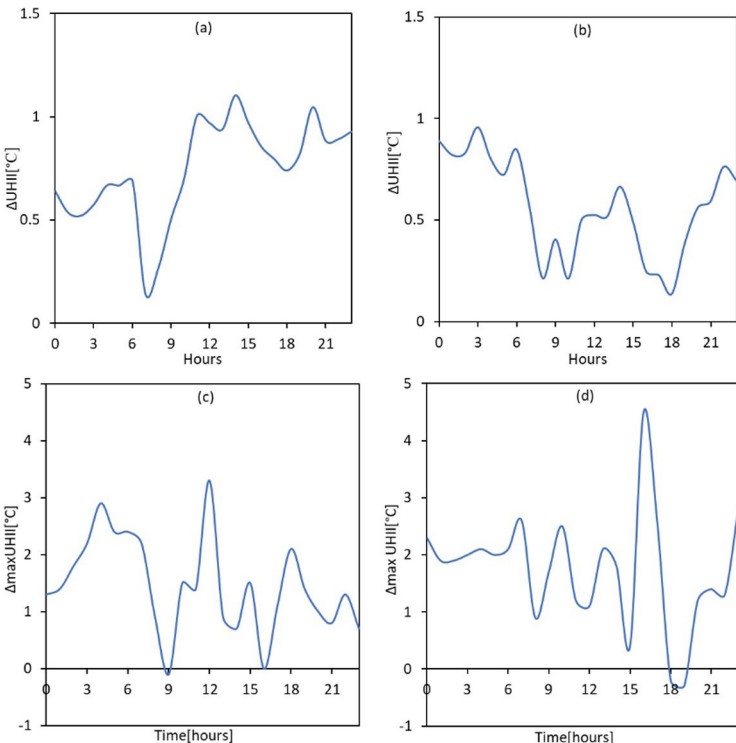

**Fig 2. Diurnal differences in UHII magnitude between HW and NHW periods; average maximum UHII values.** (a) 2012 (b) 2016; maximum values (c) 2012 (d) 2016.

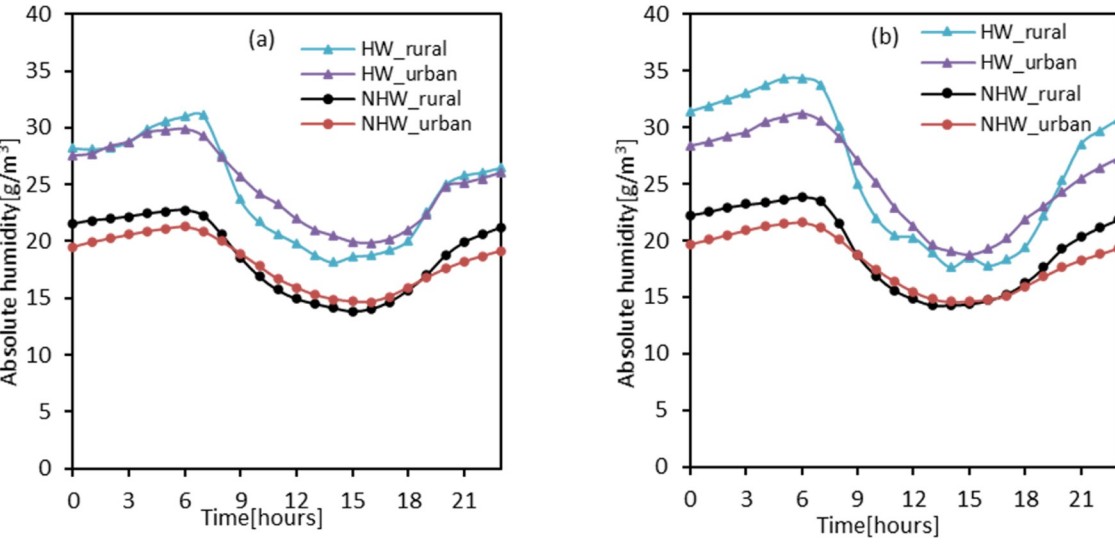

**Fig 3. Absolute humidity levels during HW and NHW periods. (a)** 2012 **(b)** 2016.

the HW period, considering all the ten candidate stations, was 29 g/m at 6 am, while the highest absolute humidity value recorded during the NHW period was 20.09 g/m$^3$ which also occurred at 6 am. Peak absolute humidity values at the rural station also occurred at 6 am and were relatively higher during the HW period (i.e., 31 g/m$^3$) than the NHW period (i.e., 22 g/m3). Similar results were observed in 2016, suggesting potential influences of relative humidity on the synergies between UHII and HWs. Also, the observed changes in absolute humidity were observed to follow a diurnal trend peaking in the early mornings before gradually decreasing, reaching their lowest in the late afternoons before gradually increasing again.

### 3.2. Wind speed, UHII and HW periods

Wind speeds ranging from 0 m/s– 13.7 m/s and 0 m/s– 10.1 m/s were observed in 2012 and 2016, respectively. In general, wind speeds tended to be low in the late night to early morning hours and started to gradually increase, reaching their peak in the afternoons before gradually decreasing again. Generally, (See S1–S10 Figs), wind speeds were relatively lower during HW periods than NHW periods considering all ten stations. In 2012, mean wind speeds during HW and NHW periods were 1.62 m/s and 1.48 m/s, respectively. In 2016, the observed mean wind speeds during HW and NHW periods were 1.21 m/s and 1.41 m/s, respectively. Again, (See S1–S10 Figs), despite a similar pattern in the diurnal variation of wind speed during both HW and NHW periods, wind speeds were more stable during the NHW period than the HW period. For instance, wind speeds during HW periods tended to fluctuate from time to time, while those in NHW periods showed a stable trend. Also, the difference in wind speeds between HW and NHW periods was much more pronounced in LCZ1, LC2 and LCZ4. To further explore the influences of wind speed on the potential synergies between HW and UHI, we studied the magnitude of UHI and the number of HW episodes under varying wind speeds: low wind speed, medium and high wind speeds. As noted in Fig 4, the number of HW is significantly higher during low wind speeds than medium and high wind speeds. Based on our data and considering all stations in 2012, we estimated 5, 11 and 12 HW periods under high, medium and low-speed conditions, respectively. Similarly, in 2016, the estimated number of HW under low wind speeds, medium wind speeds and high wind speeds were 2, 26 and 30, respectively (Fig 5).

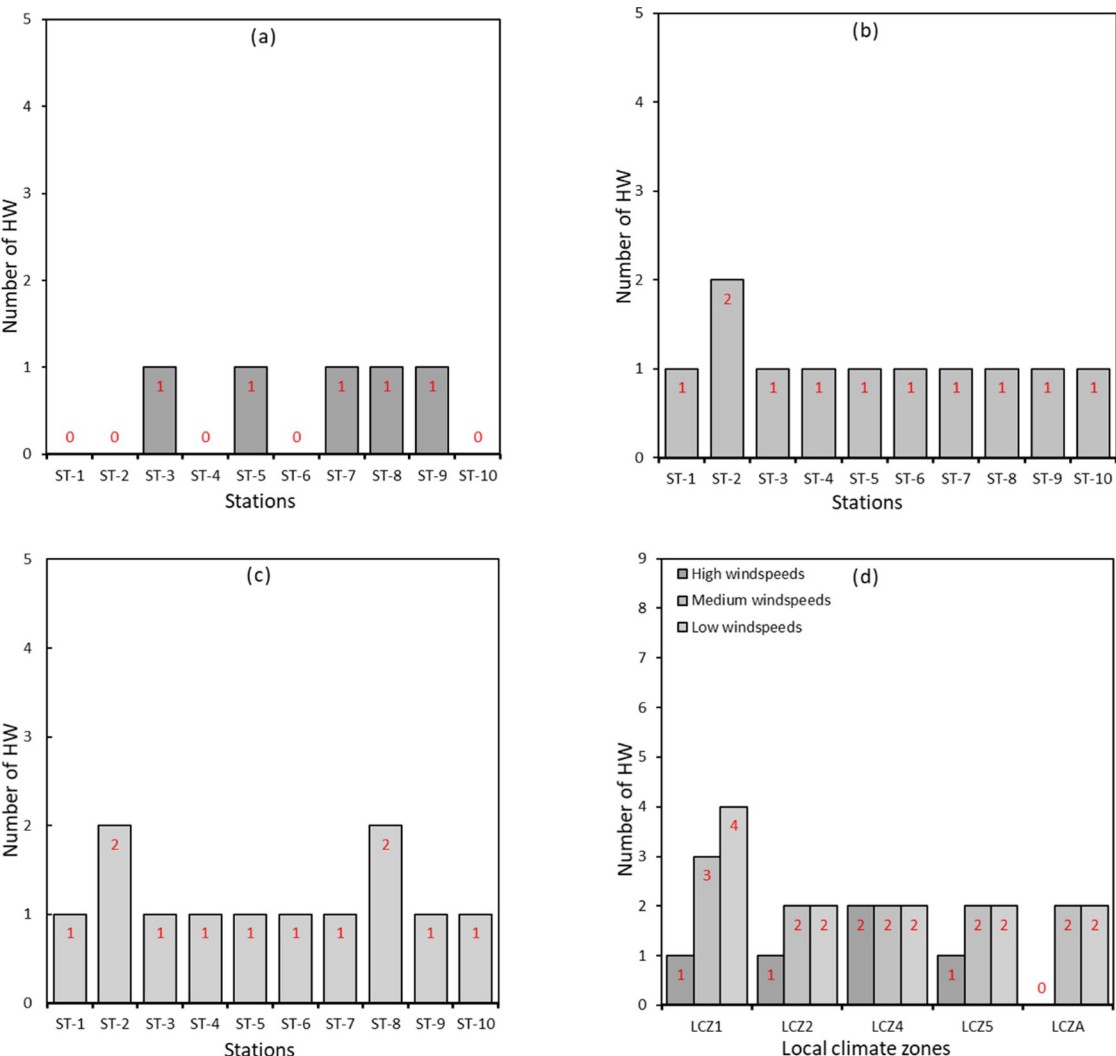

**Fig 4. Frequency of HW periods under different wind speed conditions in 2012.** (**a**) high wind speeds (**b**) medium wind speeds (**c**) low wind speeds (**d**) wind speed per LCZ.

Moreover, some stations experienced a higher number of HW periods suggesting a possible combined effect of synoptic weather conditions and land use on the frequency of HW episodes. This relationship is explored further and as seen in Figs 4D and 5D, the highest number of HW was observed under low wind speeds at stations in the LCZ1 category in both 2012 (i.e., 4 HW episodes) and 2016 (i.e., 6 HW episodes). This analysis suggests that the frequency of HW periods is amplified under conditions of low wind speeds in densely built areas.

Furthermore, mean UHII was significantly higher under conditions of low wind speeds, as seen in Fig 6. Based on the ANOVA analysis, these differences in mean UHII across the three wind speed groups were statistically significant; in 2012, F (2,36717) = 268.378, p-value < 0.001. A follow up post-hoc analysis also showed that mean UHII values in low wind speed conditions were approximately 0.70°C and 0.30°C than UHII values in high wind speeds and medium wind speeds, respectively. Similar results were observed for the year 2016, F (2,36717) = 145.511, p-value < 0.001 and a follow up post-hoc analysis showed that mean UHII values in low wind speed conditions were approximately 0.65°C and 0.195°C higher than mean UHII values under high wind speed and medium wind speeds respectively.

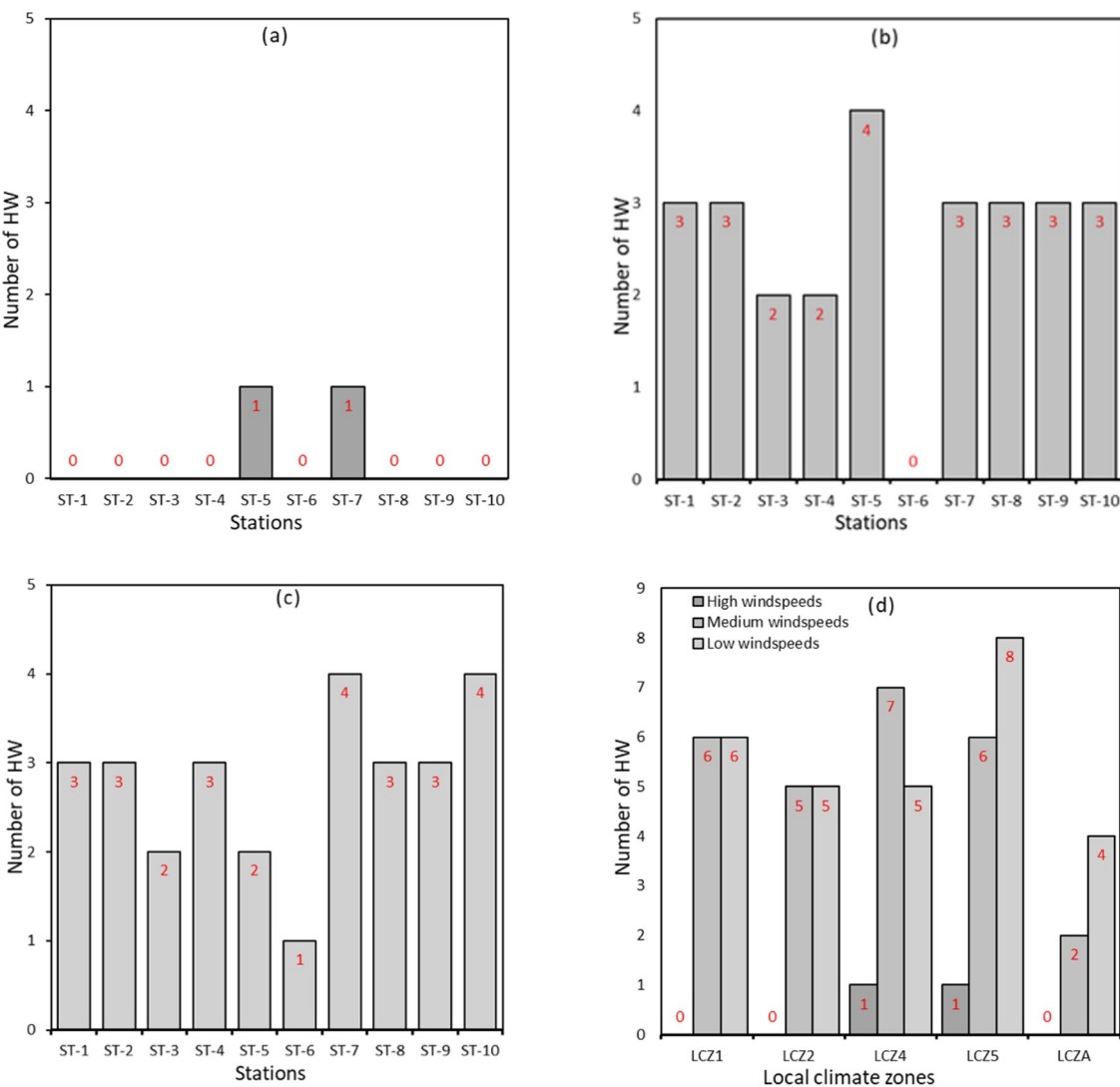

**Fig 5. Frequency of HW periods under different wind speed conditions in 2016. (a)** high wind speeds **(b)** medium wind speeds **(c)** low wind speeds **(d)** wind speed per LCZ.

### 3.3. LCZ, UHI and HW

Based on the local climate categories by Stewart and Oke [31] discussed in section 2.3, we explored differences in the number and duration of HW episodes among the LCZ's. We found that the number of HW was highest in LCZ1 in both 2012 (i.e., four episodes) and 2016 (I.e., six episodes). Conversely, the minimum number of HW episodes was observed in LCZ4, LCZ 5, LCZ A in 2012 and LCZ A in 2016. Furthermore, we explored the duration of HW in each LCZ. We found that the total number of HW days was much higher in LCZ1 than the other LCZs. For example, in 2012, the total number of days that experienced HW was 24 in LCZ1 and 15 in LCZA. Similar numbers were observed in 2016 with 60 days in LCZ1 and 21 days in LCZA. This finding suggests that the duration HW episodes are likely to be particularly long in conditions that favor intensified UHII levels, such as densely built areas with structures extending above the ground level. Fig 7 shows the frequency and duration of HWs categorized by LCZs in 2012 and 2016.

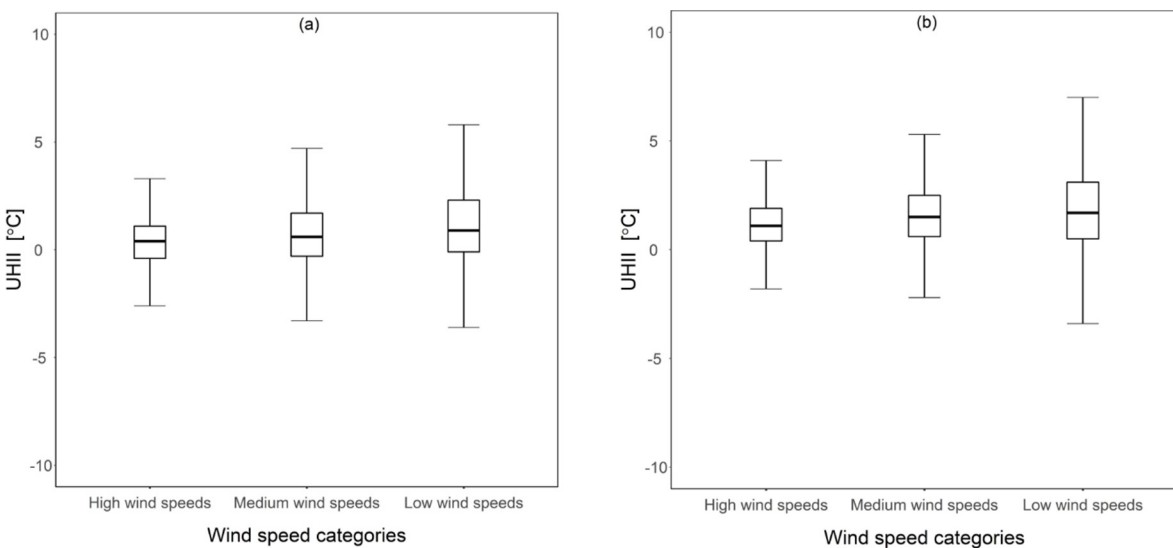

**Fig 6. Mean UHII values under different wind speed conditions. (a)** 2012 **(b)** 2016.

Furthermore, as indicated in Fig 8, the highest mean UHI levels were recorded at stations found in LCZ1. The difference in mean UHI levels across different LCZs was tested using ANOVA and found to be statistically significant. In 2012, F (4,36715) = 1817.084, p-value < 0.001. Also, a follow up post-hoc analysis showed that mean UHII values in LCZ 1 were 0.20˚C, 0.27˚C, 0.01˚C and 1.77˚C higher than mean UHII values in LCZ 2, LCZ 4, LCZ 5 and LCZ A respectively. Differences in mean UHII values were also statistically significant in the year 2016; F (4,36715) = 2982.132, p-value < 0.001. Similarly mean UHII values in LCZ 1 were approximately 0.60˚C, 0.55˚C, 0.41˚C and 2.49˚C higher than UHII values in LCZ 2, LCZ 4, LCZ 5 and LCZ A respectively. It is important to note that LCZ1 and LCZ2 are associated with higher building surface fractions than LCZ4, LCZ5 and LCZA, suggesting that both increased magnitudes of UHI and HW occurrence are strongly dependent on local land features as briefly discussed in section 3.1 above.

Furthermore, we assessed wind speed levels across the various LCZ to determine the differing wind speeds as a potential cause for the varying UHII levels shown in Fig 8. Fig 9 shows the distribution of wind speed in various LCZs in both 2012 and 2016. As shown in the figure, in 2012, the median value, which is often close to the mean value tended to vary from one LCZ to another with the highest median value observed in LCZ 4 (1.7 m/s) and the lowest in LCZA (1 m/s). Similar trends were observed for the year 2016. The trends in Fig 9 can be compared to that in Fig 8, which shows the median UHII values per LCZ. By comparing the figures, it is observed that the LCZs with the highest median UHI tended to have the lowest wind speed values and vice-versa. Surprisingly, LCZA associated with less building clutters than the other LCZs showed the lowest median wind speed values among all the LCZs.

## 4. Discussion

We explore the potential effect of UHI on the intensification of HW periods in Seoul. Additionally, given the well-known effects of synoptic weather conditions and land-use characteristics on overheating in urban cities [52–54], we assess the potential influences of wind speed and land use patterns on the synergistic interactions between UHI and HW. We found that UHII is more pronounced under HW periods than under NHW periods (i.e., regular

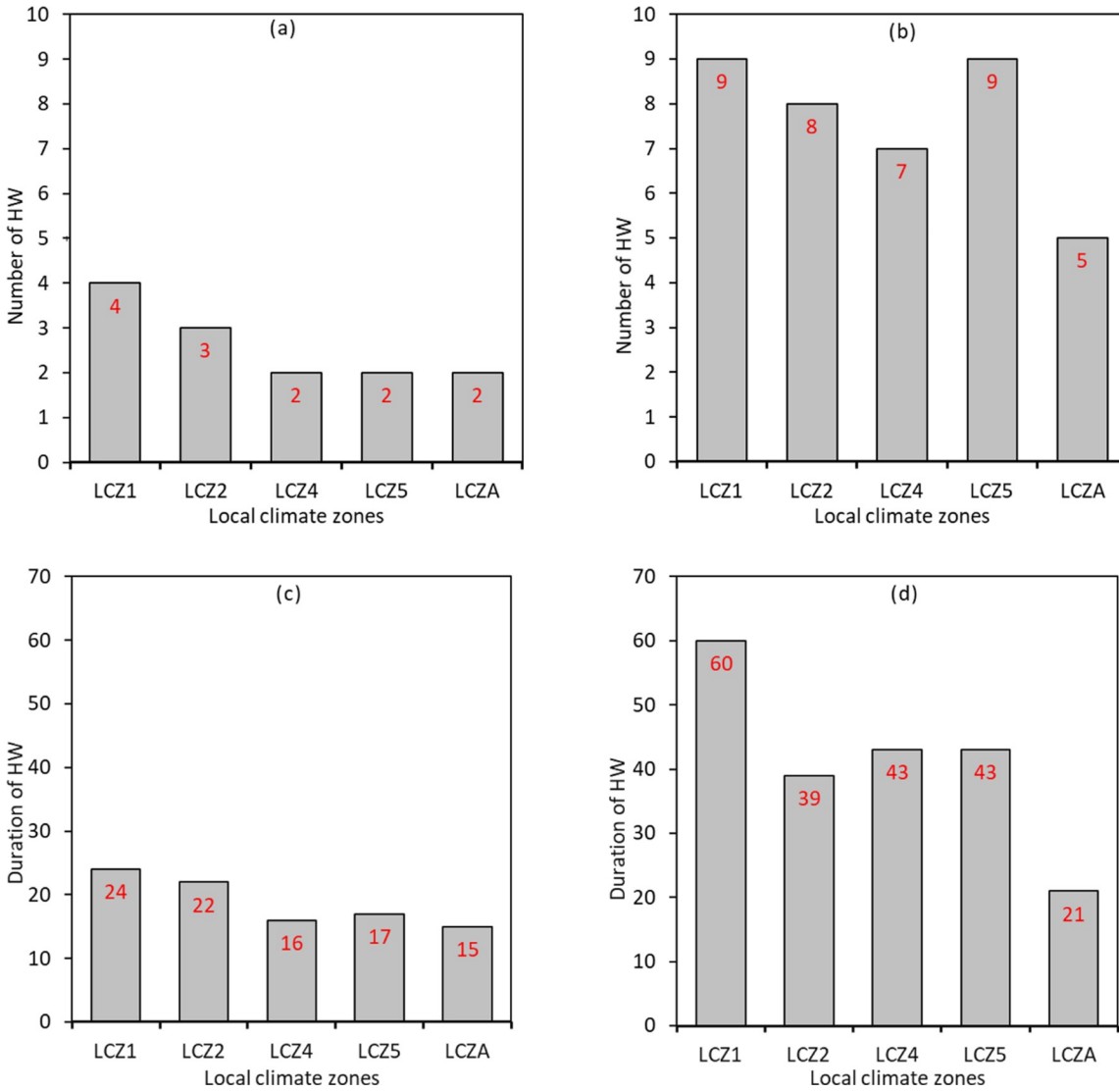

**Fig 7. Frequency of HW periods per LCZ. (a)** 2012 **(b)** 2016 and duration of HW periods per LCZ; **(c)** 2012 **(d)** 2016.

summertime conditions). While the physical mechanisms that promote intensified UHI during HW periods are not definitive and still open to scientific debate, there have been two physical processes that might explain the observed interactions between UHI and HW.

The first mechanism is related to wind speed levels associated with HW and the inherent reductive effect of wind speed on UHI magnitudes; HWs resulting from high-pressure anticyclones are often associated with low wind speeds. Similarly, low wind speeds reduce horizontal advective cooling resulting in increased UHI magnitudes. Therefore, low wind speed is a common denominator for intense HW and high UHI magnitudes, making it a substantial factor likely to influence the interactions between the two phenomena (i.e., UHI and HW). This influence of wind speed is further evidenced by the higher wind speeds experienced during the NHW periods than during the HW periods (See S1–S10 Figs). Also, we found an increased number of HW occurrences under low wind speed conditions. At the same time, we found statistically significant differences between UHII during low wind speed conditions and UHII

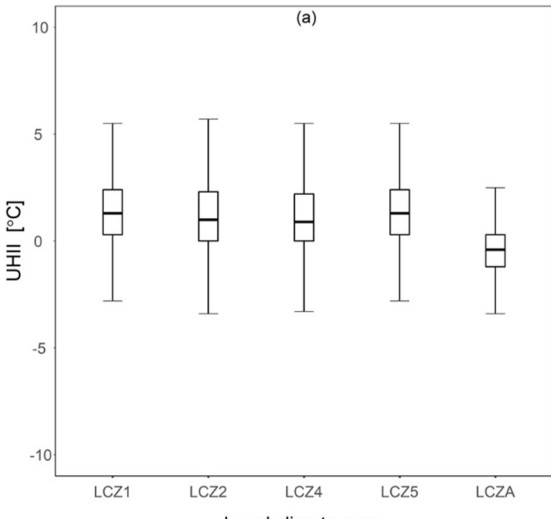
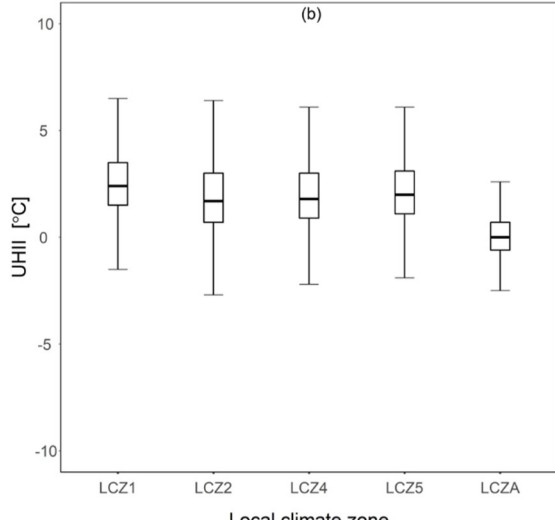

**Fig 8. Mean UHII values under different LCZs. (a)** 2012 **(b)** 2016.

during high wind speed conditions (i.e., higher UHII during low wind speed conditions than medium and high wind speed conditions). This finding suggests that the frequency of HWs and the intensity of UHI both flourish under the same synoptic weather conditions, notably reduced wind speeds are the likely reason for the differences in UHI levels between HW and NHW periods observed in Seoul city. The inner areas of Seoul, especially those categorized in LCZ 1 –LCZ 5, are characterized by large, tall buildings that hinder natural wind flow mechanisms resulting in reduced wind speeds and consequently reduced heat dissipation. Such conditions are a potential cause for the intensified UHI under HW periods and our findings reiterate previous reports [55, 56] which show relatively higher temperatures in areas with clustered building structures than those with largely sparse building structures; this is possibly as a result of wind flow obstructions by the said structures which inherently results in poor/insufficient urban ventilation [57].

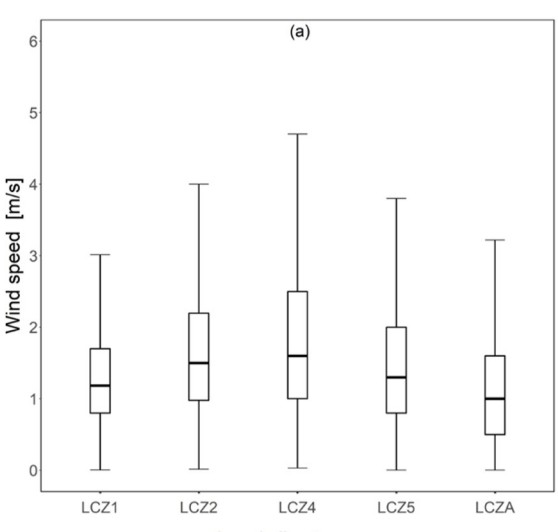
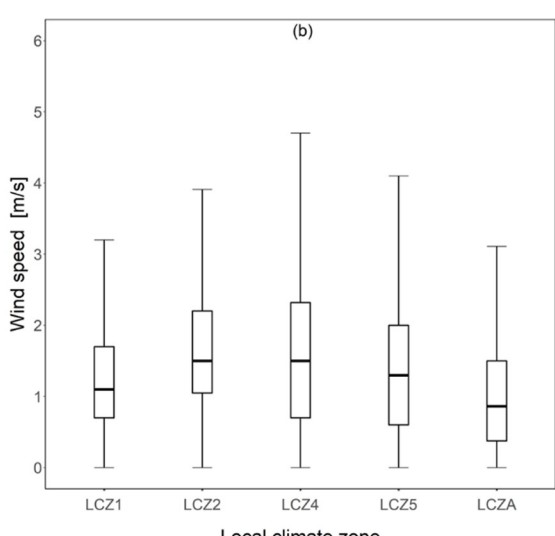

**Fig 9. Wind speed distributions across different LCZs. (a)** 2012 **(b)** 2016.

While it is impractical to artificially influence wind flow for large scale areas as a solution to reduce the synergistic effect of UHI and HW on the well-being of urban residents except maybe through long-term urban planning projects, for instance through the proper design of urban ventilation corridors for heat dispersion [58], the results provided here especially regarding the diurnal variations of the synergies between UHI and HW are of significant implications regarding when to concentrate emergency measures during extreme heat events.

Also, it is important to note that Contrary to theoretical expectations, LCZA, which is associated with low building levels that are mostly sparse and therefore likened to a rural area, tended to experience lower wind speeds than the other LCZs with tall and congested infrastructure. This finding particularly contradicts numerous studies that report lower wind speeds in urban areas than rural areas due to obstructions from tall and rough urban elements that distort natural wind flows [55–57]. One possible explanation for such a result is the wind flow regimes in dense areas with clustered buildings, which often result in the channeling / Venturi effect. In such cases, when the prevailing winds are parallel to the canyon axis, they are forced in and between clustered elements resulting in higher wind pressures and, subsequently, increased wind speeds.

Secondly, the higher UHII during HW periods than NHW periods can also be partly attributed to land use characteristics in a given area. Surface temperatures are predominantly higher during HW periods than NHW periods. Under natural surface conditions such as bare/vegetated soils found in rural areas, increased surface temperature increase evapotranspiration and thus prompting a cooling effect [30]. However, there is reduced surface moisture content in urbanized areas due to land cover materials (e.g., concrete) resulting in reduced evapotranspiration and, consequently, less evaporative cooling compared to rural areas. Moreover, artificial materials prevalent in urban areas often have increased absorptance of short wave solar radiation in the early mornings, which is later released in the form of long-wave radiation in the late afternoons to evenings. The potential impact of land use on the synergies between UHI and HW is supported by our results that show increased HW duration and frequencies as well as pronounced UHI intensifications under densely built areas such as those categorized under LCZ1—LCZ5 and relatively low HW frequencies and low UHI levels under LCZ A (S6 Fig).

Moreover, the difference between UHII during HW and NHW periods differed among LCZs. LCZs associated with high building density often showing more intensified UHII differences between HW and NHW periods than LCZs associated with low building densities (i.e., LCZ A) (See S1–S10 Figs). This is an important finding especially in regards to which areas in Seoul city are considered priorities for emergency relief during HW periods: densely built areas should be given priority as they are likely to experience a combined effect of UHI and HW thus exacerbating the health risks to demographics residing in these areas. In some instances, however, areas characterized by high building densities but in close proximity to the Han river, for example, ST-5, showed less intense synergies than those characterized by low building densities. This finding suggests that specific UHI mitigative strategies, especially those that behave as heat sinks (e.g., artificial water bodies), are likely to minimize the observed interactions between UHI and HW and thus reducing heat stress-related health risks on urban dwellers. While the idea of minimizing the combined effect of UHI and HW on urban dwellers through UHI mitigative strategies is rather self-explanatory and backed by extensive scientific reports [59–62], it is not clear which UHI mitigative measures are likely to provide optimum solutions. Specific techniques such as the use of water irrigation are likely to modify surface latent flux through increased evapotranspiration and thus significantly and positively influencing the synergies between UHI and HW. However, extensive studies dealing with such methods are rather rare in the literature [63]. It is, therefore, important that future studies, through a combination of simulation studies and field experiments, explore and quantify the capability

of diverse UHI mitigative and adaptive measures to reduce the synergistic interactions between UHI and HW.

Also, the observed higher AH between HW and NHW period is consistent with previous studies [27] and is attributed to the increased temperature during HW periods that enhance evapotranspiration mechanisms, ultimately resulting in increased atmospheric water content during HW periods. At the same time, AH levels were relatively higher at the rural station than the urban stations during both HW and NHW periods. This finding is also consistent with previous studies and is primarily attributed to the higher moisture content in rural ground surfaces than urban ground surfaces, which are often impermeable [64–66]. The sharp increase in AH at 5 am is attributed to the evaporation of dew formed during the night, which then fully transpires by around 7 am and thus the gradual drop in AH [66].

Furthermore, we found that the frequency and duration of HW were generally higher in 2012 than in 2016. This finding seems to reiterate conclusions from theoretical studies reporting an upward trend in the frequency and duration of HWs over the years. However, the current study only considers a couple of non-consecutive years and is thus insufficient to reinforce this line of thought with certainty. Future studies that explore the synergies between UHI and HW using multi-year datasets are critical in the accurate prediction of the temporal manifestations and severity of HWs. Another possible limitation of our study is related to the breadth of the synoptic weather elements considered; we only considered the potential influence of wind speed and relative humidity on the interactions between UHI and HW. However, there are several weather elements (e.g., cloud cover and height, precipitation etc.,) with profound impacts on UHI and which are likely to influence the synergistic interactions between UHI and HW. Future studies should, therefore, explore the observed synergies between UHI and HW considering multiple weather variables.

## 5. Conclusions

HW are becoming common occurrences in many areas as a result of the changing climate. The risks, especially health risks posed by HW, are more intense in urbanized, large metropolitan areas than rural areas partly due to the additive effect of UHI. We explored the synergistic interactions between HW and UHI in Seoul city. Our results showed that UHI was more intense during HW periods than during NHW periods (i.e., regular summer background conditions). The difference in maximum peak UHII between HW and NHW was found to be 4.50˚C and is consistent with reports from previous studies conducted in non-coastal cities. We also found that the synergies between UHI and HW were influenced by land use characteristics and synoptic weather conditions. They were, for example, generally more intense in areas characterized by densely built structures and low wind speed conditions. The said synergies were also typically more intense in the afternoon to late-night hours than morning hours. Our results are helpful to policymakers and urban designers when devising HW preventive measures. They are also of importance to emergency relief providers as a resource allocation guideline, for instance, regarding which areas and time of the day to prioritize during HW periods in Seoul.

## Supporting information

**S1 Fig. UHII and wind speed during HW and NHW periods at ST-1. (a)** UHII_2012 **(b)** wind speed_2012 **(c)** UHII_2016 **(d)** wind speed_2016.
(TIF)

**S2 Fig. UHII and wind speed during HW and NHW periods at ST-2. (a)** UHII_2012 **(b)** wind speed_2012 **(c)** UHII_2016 **(d)** wind speed_2016.
(TIF)

**S3 Fig. UHII and wind speed during HW and NHW periods at ST-3. (a)** UHII_2012 **(b)** wind speed_2012 **(c)** UHII_2016 **(d)** wind speed_2016.
(TIF)

**S4 Fig. UHII and wind speed during HW and NHW periods at ST-4. (a)** UHII_2012 **(b)** wind speed_2012 **(c)** UHII_2016 **(d)** wind speed_2016.
(TIF)

**S5 Fig. UHII and wind speed during HW and NHW periods at ST-5. (a)** UHII_2012 **(b)** wind speed_2012 **(c)** UHII_2016 **(d)** wind speed_2016.
(TIF)

**S6 Fig. UHII and wind speed during HW and NHW periods at ST-6. (a)** UHII_2012 **(b)** wind speed_2012 **(c)** UHII_2016 **(d)** wind speed_2016.
(TIF)

**S7 Fig. UHII and wind speed during HW and NHW periods at ST-7. (a)** UHII_2012 **(b)** wind speed_2012 **(c)** UHII_2016 **(d)** wind speed_2016.
(TIF)

**S8 Fig. UHII and wind speed during HW and NHW periods at ST-8. (a)** UHII_2012 **(b)** wind speed_2012 **(c)** UHII_2016 **(d)** wind speed_2016.
(TIF)

**S9 Fig. UHII and wind speed during HW and NHW periods at ST-9. (a)** UHII_2012 **(b)** wind speed_2012 **(c)** UHII_2016 **(d)** wind speed_2016.
(TIF)

**S10 Fig. UHII and wind speed during HW and NHW periods at ST-10. (a)** UHII_2012 **(b)** wind speed_2012 **(c)** UHII_2016 **(d)** wind speed_2016.
(TIF)

## Author Contributions

**Conceptualization:** Mat Santamouris, Geun Young Yun.

**Data curation:** Jacques Nganyiyimana.

**Formal analysis:** Jack Ngarambe, Jacques Nganyiyimana, Mat Santamouris, Geun Young Yun.

**Funding acquisition:** Inhan Kim, Geun Young Yun.

**Methodology:** Mat Santamouris, Geun Young Yun.

**Supervision:** Geun Young Yun.

**Validation:** Inhan Kim, Geun Young Yun.

**Visualization:** Jacques Nganyiyimana.

**Writing – original draft:** Jack Ngarambe, Geun Young Yun.

**Writing – review & editing:** Jack Ngarambe, Mat Santamouris.

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
