## [Decision Letter · Decision Letter 0]

20 Aug 2020

PONE-D-20-24110

Synergies between urban heat island and heat waves in Seoul: The role of wind speed and land use characteristics

PLOS ONE

Dear Dr. Yun,

Thank you for submitting your manuscript to PLOS ONE. After careful consideration, we feel that it has merit but does not fully meet PLOS ONE’s publication criteria as it currently stands. Therefore, we invite you to submit a revised version of the manuscript that addresses the points raised during the review process.

We look forward to receiving your revised manuscript.

Kind regards,

Jun Yang

Academic Editor

PLOS ONE

Journal Requirements:

2. Please amend the manuscript submission data (via Edit Submission) to include author Mat Santamouris.

3. We note that Figure 1 in your submission contain map/satellite images which may be copyrighted. All PLOS content is published under the Creative Commons Attribution License (CC BY 4.0), which means that the manuscript, images, and Supporting Information files will be freely available online, and any third party is permitted to access, download, copy, distribute, and use these materials in any way, even commercially, with proper attribution. For these reasons, we cannot publish previously copyrighted maps or satellite images created using proprietary data, such as Google software (Google Maps, Street View, and Earth). For more information, see our copyright guidelines: http://journals.plos.org/plosone/s/licenses-and-copyright.

a) You may seek permission from the original copyright holder of Figure(s) [#] to publish the content specifically under the CC BY 4.0 license.

Additional Editor Comments (if provided):

Reviewer 1

Thank you for providing me the opportunity to review the paper by Ngarambe et al. on the urban heat island in Seoul. I enjoyed the piece and I found the paper to be well organized and very well written. The key references are cited, the abstract is informative, the length is fine, the methods are appropriate and innovative, the graphics are publishable (the graphics are a strong component to this paper), and the results are interesting. I found the entire effort to be of the highest quality.

Reviewer 2

This should be an interesting paper which aims to see the synergies between heat waves and urban heat island, as well as the impacts of wind speed and land use characteristics (local climate zones). However, the importance of wind speed and land use characteristics has not been well presented and analysed.

First of all, I believe, it should be wind speed rather than windspeed.

Introduction

Line 48-71, One-page description of the HW-related death is too long. This has been well known. Do not need to list so many cases. Cut down.

Line 89-90, references. Can you enumerate some irregular cases?

Line 106-112, Is it a robust cause or research gap that motivates you to analyse the synergistic interaction in Seoul? The research gap is weak.

Please double check if the word spelling ‘synergistic’ or ‘synergetic’

Authors have just mentioned the synergies between heat wave and UHI, as well as the local climate zone by Oke. However, the wind speed thing has not been given at the local precinct scale. Please also include the wind impact on the local warming mitigation. I have read some papers about the wind cooling potential, in the open low-rise gridiron precinct, open midrise gridiron precinct and compact high-rise gridiron precinct published recently, self-search online. Such wind literature can help explain, why you will start from the wind speed and land use characteristics.

Methods:

Please indicate the location of ten weather stations and surrounding building conditions. As I know, the wind and temperature recorded by the national weather stations can also be affected by the surrounding buildings.

Table 1: unit

Line 253, p not P

Results:

Figure 2-11 should be moved to the appendix

Figure 16 and Figure 18 you should also do the comparative analysis.

Discussion

Line 470-502, remove, just a repetition of the introduction and results, not the discussion

Line 519-523, your discussion should be supported by the references.

See papers: Influences of urban spatial form on urban heat island effects at the community level in China, Sustainable Cities and Society (2020), doi: https://doi.org/10.1016/j.scs.2019.101972

Spatial differentiation of urban wind and thermal environment in different grid sizes. Urban Climate 28(2019):1-13. https://doi.org/10.1016/j.uclim.2019.100458.

Local Climate Zone Ventilation and Urban Land Surface Temperatures: Towards a Performance-based and Wind-sensitive Planning Proposal in Megacities. Sustainable Cities and Society, 47(2019):1-11.DOI:10.1016/j.scs.2019.101487.

Also: there is a paper about the local climate zone and local-scale urban ventilation performance published in the journal of Atmosphere. You may read it.

Reviewers' comments:

Reviewer's Responses to Questions

**Comments to the Author**

1. Is the manuscript technically sound, and do the data support the conclusions?

Reviewer #1: Yes

Reviewer #2: Partly

2. Has the statistical analysis been performed appropriately and rigorously? 

Reviewer #1: Yes

Reviewer #2: No

3. Have the authors made all data underlying the findings in their manuscript fully available?

Reviewer #1: Yes

Reviewer #2: Yes

4. Is the manuscript presented in an intelligible fashion and written in standard English?

Reviewer #1: Yes

Reviewer #2: Yes

5. Review Comments to the Author

Reviewer #1: Thank you for providing me the opportunity to review the paper by Ngarambe et al. on the urban heat island in Seoul. I enjoyed the piece and I found the paper to be well organized and very well written. The key references are cited, the abstract is informative, the length is fine, the methods are appropriate and innovative, the graphics are publishable (the graphics are a strong component to this paper), and the results are interesting. I found the entire effort to be of the highest quality.

Reviewer #2: Major revision

This should be an interesting paper which aims to see the synergies between heat waves and urban heat island, as well as the impacts of wind speed and land use characteristics (local climate zones). However, the importance of wind speed and land use characteristics has not been well presented and analysed.

First of all, I believe, it should be wind speed rather than windspeed.

Introduction

Line 48-71, One-page description of the HW-related death is too long. This has been well known. Do not need to list so many cases. Cut down.

Line 89-90, references. Can you enumerate some irregular cases?

Line 106-112, Is it a robust cause or research gap that motivates you to analyse the synergistic interaction in Seoul? The research gap is weak.

Please double check if the word spelling ‘synergistic’ or ‘synergetic’

Authors have just mentioned the synergies between heat wave and UHI, as well as the local climate zone by Oke. However, the wind speed thing has not been given at the local precinct scale. Please also include the wind impact on the local warming mitigation. I have read some papers about the wind cooling potential, in the open low-rise gridiron precinct, open midrise gridiron precinct and compact high-rise gridiron precinct published recently, self-search online. Such wind literature can help explain, why you will start from the wind speed and land use characteristics.

Methods:

Please indicate the location of ten weather stations and surrounding building conditions. As I know, the wind and temperature recorded by the national weather stations can also be affected by the surrounding buildings.

Table 1: unit

Line 253, p not P

Results:

Figure 2-11 should be moved to the appendix

Figure 16 and Figure 18 you should also do the comparative analysis.

Discussion

Line 470-502, remove, just a repetition of the introduction and results, not the discussion

Line 519-523, your discussion should be supported by the references.

See papers: Influences of urban spatial form on urban heat island effects at the community level in China, Sustainable Cities and Society (2020), doi: https://doi.org/10.1016/j.scs.2019.101972

Spatial differentiation of urban wind and thermal environment in different grid sizes. Urban Climate 28(2019):1-13. https://doi.org/10.1016/j.uclim.2019.100458.

Local Climate Zone Ventilation and Urban Land Surface Temperatures: Towards a Performance-based and Wind-sensitive Planning Proposal in Megacities. Sustainable Cities and Society, 47(2019):1-11.DOI:10.1016/j.scs.2019.101487.

Also: there is a paper about the local climate zone and local-scale urban ventilation performance published in the journal of Atmosphere. You may read it.

6. PLOS authors have the option to publish the peer review history of their article (what does this mean?). If published, this will include your full peer review and any attached files.

Reviewer #1: No

Reviewer #2: No

---

## [Author Response · Author response to Decision Letter 0]

18 Nov 2020

EDITOR COMMENTS

Editor comment 1

Authors' response

We have ensured that the manuscript adheres to the style requrements instructed by PLOS ONE.

Editor comment 2

Please amend the manuscript submission data (via Edit Submission) to include author Mat Santamouris.

Authors' response

We have amended the submission data to include author Mat Santamouris

Editor comment 3

We note that Figure 1 in your submission contain map/satellite images which may be copyrighted. All PLOS content is published under the Creative Commons Attribution License (CC BY 4.0), which means that the manuscript, images, and Supporting Information files will be freely available online, and any third party is permitted to access, download, copy, distribute, and use these materials in any way, even commercially, with proper attribution. For these reasons, we cannot publish previously copyrighted maps or satellite images created using proprietary data, such as Google software (Google Maps, Street View, and Earth). For more information, see our copyright guidelines: http://journals.plos.org/plosone/s/licenses-and-copyright.

Authors' response

We have remade the figure using an alternative platform,openmap, which is free and requires no particular permission (https://www.openstreetmap.org/user/new?cookie_test=true)

Editor comment 4

Please include captions for your Supporting Information files at the end of your manuscript, and update any in-text citations to match accordingly. Please see our Supporting Information guidelines for more information: http://journals.plos.org/plosone/s/supporting-information.

Authors' response

 We have added the captions for the supporting information files

REVIEWER COMMENTS

Reviewer comment 1

First of all, I believe, it should be wind speed rather than windspeed. 

Authors response

Thank you for the comment. We have corrected the terminology throughout the manuscript.

Reviewer comment 2

Line 48-71, One-page description of the HW-related death is too long. This has been well known. Do not need to list so many cases. Cut down. 

Authors' response

We have shortened this part of the manuscript as below.

Modified text (Line 59 – Line 68)

“HWs have also been reported to significantly contribute to the incessant number of heat-related mortalities around the world; the 2003 European HWs, which resulted in approximately 70,000 deaths, is a good example of the extreme fatal risk posed by HWs [6]. Relatively less severe but equally high numbers of HW-related mortalities have been reported in the united states [7], Russia [8] and Korea [9]. Moreover, studies have shown that global warming-related increase in temperature is likely to exacerbate both the intensity and frequency of HW in many parts of the world. Also, through dynamic modeling and simulation studies, there is strong evidence that the occurrence, severity and duration of HW periods are also likely to increase in the near future [10,11]”

References

6. Robine J-M, Cheung SLK, Le Roy S, Van Oyen H, Griffiths C, Michel J-P, et al. Death toll exceeded 70,000 in Europe during the summer of 2003. Comptes Rendus Biologies. 2008;331: 171–178. doi:10.1016/j.crvi.2007.12.001

7. Knowlton K, Rotkin-Ellman M, King G, Margolis HG, Smith D, Solomon G, et al. The 2006 California Heat Wave: Impacts on Hospitalizations and Emergency Department Visits. Environmental Health Perspectives. 2009;117: 61–67. doi:10.1289/ehp.11594

8. Barriopedro D, Fischer EM, Luterbacher J, Trigo RM, Garcia-Herrera R. The Hot Summer of 2010: Redrawing the Temperature Record Map of Europe. Science. 2011;332: 220–224. doi:10.1126/science.1201224

9. Choi N, Lee M-I. Spatial Variability and Long-Term Trend in the Occurrence Frequency of Heatwave and Tropical Night in Korea. Asia-Pacific J Atmos Sci. 2019;55: 101–114. doi:10.1007/s13143-018-00101-w

10. Perkins SE, Alexander LV, Nairn JR. Increasing frequency, intensity and duration of observed global heatwaves and warm spells. Geophys Res Lett. 2012;39: 2012GL053361. doi:10.1029/2012GL053361

11. Meehl GA. More Intense, More Frequent, and Longer Lasting Heat Waves in the 21st Century. Science. 2004;305: 994–997. doi:10.1126/science.1098704

Reviewer comment 3

Line 89-90, references. Can you enumerate some irregular cases? 

Authors' response

We have modified the text to include some references that validate the statements.

Modified text (Line 90 – Line 96)

“For instance, Yang et al. [20] report a 1.9°C difference in surface temperature between American metropolitan cities and their rural counterparts during HW periods. Similarly, Tan et al. [21] observed higher degrees of warming in Shanghai relative to its surrounding area during HW periods. Moreover, heat-related mortalities during HW periods have been observed more in urban areas than in rural areas [22,23]. In essence, therefore, urban dwellers are more vulnerable to heat-stress-related health conditions than rural dwellers as they are likely to combat HWS and UHI's synergistic effects.”

References

20. Yang J, Hu L, Wang C. Population dynamics modify urban residents’ exposure to extreme temperatures across the United States. Sci Adv. 2019;5: eaay3452. doi:10.1126/sciadv.aay3452

21. Tan J, Zheng Y, Tang X, Guo C, Li L, Song G, et al. The urban heat island and its impact on heat waves and human health in Shanghai. Int J Biometeorol. 2010;54: 75–84. doi:10.1007/s00484-009-0256-x

22. Rydman RJ, Rumoro DP, Silva JC, Hogan TM, Kampe LM. The Rate and Risk of Heat-Related Illness in Hospital Emergency Departments During the 1995 Chicago Heat Disaster. : 16.

23. Grize L, Huss A, Thommen O, Schindler C, Braun-Fahrländer C. Heat wave 2003 and mortality in Switzerland. SWISS MED WKLY. 2003; 7.

Reviewer comment 4

Line 106-112, Is it a robust cause or research gap that motivates you to analyse the synergistic interaction in Seoul? The research gap is weak. 

Authors' response

We regret that our reasons for undertaking this study were unclear in the original manuscript. We have succinctly modified the text for clarity. Ideally, while the manuscript sheds more light on the interaction between UHI and HWs in light of the ongoing climate change dynamics, we particularly hoped to quantify the synergies between UHI and HWs for Seoul city. Our primary motivation for doing such is because, while there are several studies on the synergies between UHI and HW in several metropolitan cities, their results are unlikely to be accurately extrapolated to Seoul city, given the obvious differences in terms of topography, climate, population, etc. 

We have re-written our motivation for conducting this study more clearly. Please see the modified text below

Modified text (Line 112 – Line 121)

“While the synergistic interactions between UHI and HWs have been studied, to an extent, in some large cities around the world which are discussed above [25–28], it is essential to remark that the synergies between UHI and HW are likely to be influenced by factors (e.g., synoptic weather, topography, population density, land use) specific to a given region/city. As such, the synergies between UHI and HW within a given area are distinctive rather than general and must be quantified for individual areas of interest. To that end, we attempt to explore the potential synergistic interactions between UHI and HW in Seoul, a large metropolitan city with a dense population, advanced urbanization, which inherently suffers from the urban warming-related problems discussed above.”

References

25. Founda D, Santamouris M. Synergies between Urban Heat Island and Heat Waves in Athens (Greece), during an extremely hot summer (2012). Sci Rep. 2017;7: 10973. doi:10.1038/s41598-017-11407-6

26. Khan HS, Paolini R, Santamouris M, Caccetta P. Exploring the Synergies between Urban Overheating and Heatwaves (HWs) in Western Sydney. Energies. 2020;13: 470. doi:10.3390/en13020470

27. Heaviside C, Vardoulakis S, Cai X-M. Attribution of mortality to the urban heat island during heatwaves in the West Midlands, UK. Environ Health. 2016;15: S27. doi:10.1186/s12940-016-0100-9

28. Tan J, Zheng Y, Tang X, Guo C, Li L, Song G, et al. The urban heat island and its impact on heat waves and human health in Shanghai. Int J Biometeorol. 2010;54: 75–84. doi:10.1007/s00484-009-0256-x

Reviewer comment 5

Please double check if the word spelling ‘synergistic’ or ‘synergetic’

Authors' response

We have changed the terminology to “synergistic” throughout the manuscript.

Reviewer comment 6

Authors have just mentioned the synergies between heat wave and UHI, as well as the local climate zone by Oke. However, the wind speed thing has not been given at the local precinct scale. Please also include the wind impact on the local warming mitigation. I have read some papers about the wind cooling potential, in the open low-rise gridiron precinct, open midrise gridiron precinct and compact high-rise gridiron precinct published recently, self-search online. Such wind literature can help explain, why you will start from the wind speed and land use characteristics.

Authors' response

Thanks for pointing this out. We have added text to highlight the mitigation capabilities of wind speeds and consequently, its potential to alleviate the synergistic effects of UHI and HWs on urban dwellers. 

Added text (Line 128 – Line 134)

“Also, the advective cooling resulting from prevailing winds naturally reduces local warming. This heat-mitigative effect of wind speeds has been illustrated in areas of various morphologies and building structures ranging from low-rise [31] to open mid-rise gridiron precincts [32]. It has also been previously shown that above certain wind speed thresholds, UHI is fully diminished / gets close to zero [33]. We thus study the difference in UHI magnitudes between HW and NHW periods under different wind conditions.”

References

31. He B-J, Ding L, Prasad D. Urban ventilation and its potential for local warming mitigation: A field experiment in an open low-rise gridiron precinct. Sustainable Cities and Society. 2020;55: 102028. doi:10.1016/j.scs.2020.102028

32. He B-J, Ding L, Prasad D. Wind-sensitive urban planning and design: Precinct ventilation performance and its potential for local warming mitigation in an open midrise gridiron precinct. Journal of Building Engineering. 2020;29: 101145. doi:10.1016/j.jobe.2019.101145

33. He B-J. Potentials of meteorological characteristics and synoptic conditions to mitigate urban heat island effects. Urban Climate. 2018;24: 26–33. doi:10.1016/j.uclim.2018.01.004

Reviewer comment 7

Please indicate the location of ten weather stations and surrounding building conditions. As I know, the wind and temperature recorded by the national weather stations can also be affected by the surrounding buildings. 

Authors' response

We agree. The stations' locations and surrounding characteristics (i.e., building surface fraction and Height of rough elements), which we used to classify the stations into local climatic zones and simultaneously understand each station's building characteristics, are provided. Please see Table 1 and Figure 1 in the revised manuscript

Reviewer comment 8

Table1: unit 

Authors response

We have added the units

Reviewer comment 9

Line 253, p not P 

Authors' response

We have corrected the error.

Reviewer comment 10

Figure 2-11 should be moved to the appendix

Authors' response

We have moved the figures to the appendix section

Reviewer comment 11

Figure 16 and Figure 18 you should also do the comparative analysis.

Authors' response

We agree with your assessment and we have added a new figure and text comparing the wind speed levels across the different LCZ. This further emphasizes the role of wind speed in reducing UHI and subsequently, HWs. See the added text below and Figure 9 in the revised manuscript

Added text and figure (Line 447– Line 457) 

“Furthermore, we assessed wind speed levels across the various LCZ to determine the differing wind speeds as a potential cause for the varying UHII levels shown in Fig 8. Fig 9 shows the distribution of wind speed in various LCZs in both 2012 and 2016. As shown in the figure, in 2012, the median value, which is often close to the mean value tended to vary from one LCZ to another with the highest median value observed in LCZ 4 (1.7 m/s) and the lowest in LCZA (1 m/s). Similar trends were observed for the year 2016. The trends in Fig 9 can be compared to that in Figure 8, which shows the median UHII values per LCZ. By comparing the figures, it is observed that the LCZs with the highest median UHI tended to have the lowest wind speed values and vice-versa. Surprisingly, LCZA associated with less building clutters than the other LCZs showed the lowest median wind speed values among all the LCZs.”

Discussion of this particular result (Line 506 - Line 518)

“Also, it is important to note that Contrary to theoretical expectations, LCZA, which is associated with low building levels that are mostly sparse and therefore likened to a rural area, tended to experience lower wind speeds than the other LCZs with tall and congested infrastructure. This finding particularly contradicts numerous studies that report lower wind speeds in urban areas than rural areas due to obstructions from tall and rough urban elements that distort natural wind flows [54–56]. One possible explanation for such a result is the wind flow regimes in dense areas with clustered buildings, which often result in the channeling/Venturi effect. In such cases, when the prevailing winds are parallel to the canyon axis, they are forced in and between clustered elements resulting in higher wind pressures and, subsequently, increased wind speeds.

References

54. Guo A, Yang J, Xiao X, Xia (Cecilia) J, Jin C, Li X. Influences of urban spatial form on urban heat island effects at the community level in China. Sustainable Cities and Society. 2020;53: 101972. doi:10.1016/j.scs.2019.101972

55. Zhao Z, Shen L, Li L, Wang H, He B-J. Local Climate Zone Classification Scheme Can Also Indicate Local-Scale Urban Ventilation Performance: An Evidence-Based Study. Atmosphere. 2020;11: 776. doi:10.3390/atmos11080776

56. Yang J, Wang Y, Xiao X, Jin C, Xia J (Cecilia), Li X. Spatial differentiation of urban wind and thermal environment in different grid sizes. Urban Climate. 2019;28: 100458. doi:10.1016/j.uclim.2019.100458

Reviewer comment 12

Line 470-502, remove, just a repetition of the introduction and results, not the discussion

Authors' response

Thank you for the comment. We have removed the suggested text and briefly stated the study's aims and general findings before quickly moving to the actual discussion. See below the modified text.

Added text (Line 463 – Line 471)

“We explore the potential effect of UHI on the intensification of HW periods in Seoul. Additionally, given the well-known effects of synoptic weather conditions and land-use characteristics on overheating in urban cities [51–53], we assess the potential influences of wind speed and land use patterns on the synergistic interactions between UHI and HW. We found that UHII is more pronounced under HW periods than under NHW periods (i.e., regular summertime conditions). While the physical mechanisms that promote intensified UHI during HW periods are not definitive and still open to scientific debate, there have been two physical processes that might explain the observed interactions between UHI and HW.”

References

51. Morris CJG, Simmonds I, Plummer N. Quantiﬁcation of the Inﬂuences of Wind and Cloud on the Nocturnal Urban Heat Island of a Large City. JOURNAL OF APPLIED METEOROLOGY. 2001;40: 14.

52. Bokaie M, Zarkesh MK, Arasteh PD, Hosseini A. Assessment of Urban Heat Island based on the relationship between land surface temperature and Land Use/ Land Cover in Tehran. Sustainable Cities and Society. 2016;23: 94–104. doi:10.1016/j.scs.2016.03.009

53. Cheung PK, Jim CY. Effects of urban and landscape elements on air temperature in a high-density subtropical city. Building and Environment. 2019;164: 106362. doi:10.1016/j.buildenv.2019.106362

Reviewer comment 13

Line 519-523, your discussion should be supported by the references.

See papers: 

Influences of urban spatial form on urban heat island effects at the community level in China, Sustainable Cities and Society (2020), doi: https://doi.org/10.1016/j.scs.2019.101972

Spatial differentiation of urban wind and thermal environment in different grid sizes. Urban Climate 28(2019):1-13. https://doi.org/10.1016/j.uclim.2019.100458.

Local Climate Zone Ventilation and Urban Land Surface Temperatures: Towards a Performance-based and Wind-sensitive Planning Proposal in Megacities. Sustainable Cities and Society, 47(2019):1-11.DOI:10.1016/j.scs.2019.101487.

Also: there is a paper about the local climate zone and local-scale urban ventilation performance published in the journal of Atmosphere. You may read it.

Authors' response

We thank the reviewer for the references which we found helpful in supporting our findings. We have referenced the suggested studies in the modified version of the discussion. See below the modified text where the references are used. 

Added text (Line 485 – Line 504)

“This finding suggests that the frequency of HWs and the intensity of UHI both flourish under the same synoptic weather conditions, notably reduced wind speeds are the likely reason for the differences in UHI levels between HW and NHW periods observed in Seoul city. The inner areas of Seoul, especially those categorized in LCZ 1 – LCZ 5, are characterized by large, tall buildings that hinder natural wind flow mechanisms resulting in reduced wind speeds and consequently reduced heat dissipation. Such conditions are a potential cause for the intensified UHI under HW periods and our findings reiterate previous reports [54,55] which show relatively higher temperatures in areas with clustered building structures than those with largely sparse building structures; this is possibly as a result of wind flow obstructions by the said structures which inherently results in poor/insufficient urban ventilation [56]

While it is impractical to artificially influence wind flow for large scale areas as a solution to reduce the synergistic effect of UHI and HW on the well-being of urban residents except maybe through long-term urban planning projects, for instance through the proper design of urban ventilation corridors for heat dispersion [57], the results provided here especially regarding the diurnal variations of the synergies between UHI and HW are of significant implications regarding when to concentrate emergency measures during extreme heat events.”

References

54. Guo A, Yang J, Xiao X, Xia (Cecilia) J, Jin C, Li X. Influences of urban spatial form on urban heat island effects at the community level in China. Sustainable Cities and Society. 2020;53: 101972. doi:10.1016/j.scs.2019.101972

55. Zhao Z, Shen L, Li L, Wang H, He B-J. Local Climate Zone Classification Scheme Can Also Indicate Local-Scale Urban Ventilation Performance: An Evidence-Based Study. Atmosphere. 2020;11: 776. doi:10.3390/atmos11080776

56. Yang J, Wang Y, Xiao X, Jin C, Xia J (Cecilia), Li X. Spatial differentiation of urban wind and thermal environment in different grid sizes. Urban Climate. 2019;28: 100458. doi:10.1016/j.uclim.2019.100458

57. Yang J, Jin S, Xiao X, Jin C, Xia J (Cecilia), Li X, et al. Local climate zone ventilation and urban land surface temperatures: Towards a performance-based and wind-sensitive planning proposal in megacities. Sustainable Cities and Society. 2019;47: 101487. doi:10.1016/j.scs.2019.101487

---

## [Decision Letter · Decision Letter 1]

24 Nov 2020

Synergies between urban heat island and heat waves in Seoul: The role of wind speed and land use characteristics

PONE-D-20-24110R1

Dear Dr. Yun,

We’re pleased to inform you that your manuscript has been judged scientifically suitable for publication and will be formally accepted for publication once it meets all outstanding technical requirements.

Kind regards,

Jun Yang

Academic Editor

PLOS ONE

Additional Editor Comments (optional):

Accept

Reviewers' comments:

Reviewer's Responses to Questions

**Comments to the Author**

1. If the authors have adequately addressed your comments raised in a previous round of review and you feel that this manuscript is now acceptable for publication, you may indicate that here to bypass the “Comments to the Author” section, enter your conflict of interest statement in the “Confidential to Editor” section, and submit your "Accept" recommendation.

Reviewer #1: All comments have been addressed

Reviewer #2: All comments have been addressed

2. Is the manuscript technically sound, and do the data support the conclusions?

Reviewer #1: Yes

Reviewer #2: Yes

3. Has the statistical analysis been performed appropriately and rigorously? 

Reviewer #1: Yes

Reviewer #2: Yes

4. Have the authors made all data underlying the findings in their manuscript fully available?

Reviewer #1: Yes

Reviewer #2: Yes

5. Is the manuscript presented in an intelligible fashion and written in standard English?

Reviewer #1: Yes

Reviewer #2: Yes

6. Review Comments to the Author

Reviewer #1: Looks good. It appears that the authors have addressed the various suggestions of the reviewers. The paper is ready to be published.

Reviewer #2: authors have well addressed all my comments. I suggest the acceptance of this paper. Well done!

7. PLOS authors have the option to publish the peer review history of their article (what does this mean?). If published, this will include your full peer review and any attached files.

Reviewer #1: No

Reviewer #2: No

---

## [Editor Report · Acceptance letter]

27 Nov 2020

PONE-D-20-24110R1 

Synergies between urban heat island and heat waves in Seoul: The role of wind speed and land use characteristics 

Dear Dr. Yun:

I'm pleased to inform you that your manuscript has been deemed suitable for publication in PLOS ONE. Congratulations! Your manuscript is now with our production department. 

Kind regards, 

on behalf of

Dr. Jun Yang 

Academic Editor

PLOS ONE